

# Plant mercury accumulation and litter input to a Northern Sedge-dominated Peatland

Ting Sun[1,2,3] and Brian A. Branfireun[4]

[1]School of Environmental Science and Engineering, Shandong University, Qingdao 266237, PR China

[2]Institute of Eco-Environmental Forensics, Shandong University, Qingdao 266237, PR China

[3]Department of Earth Sciences, University of Western Ontario, 1151 Richmond Street, London, Ontario, Canada

[4]Department of Biology, University of Western Ontario, 1151 Richmond Street, London, Ontario, Canada

*Correspondence to*: Ting Sun (tsun64@uwo.ca)



Abstract
Plant foliage plays an essential role in accumulating mercury (Hg) from the atmosphere and transferring it to soils in terrestrial
ecosystems. While many studies have focused on forested ecosystems. Hg input from plants to northern peatland peat soils has
not been nearly as well studied and is likely equally important from a mass balance perspective. In this study, we investigated the
accumulation of atmospheric Hg by the dominant plant species, few-seeded sedge [*Carex oligosperma* Michx.], wire sedge
[*Carex lasiocarpa* Ehrh], tussock sedge [*Carex stricta* Lamb.], and sweet gale [*Myrica gale* L.] in a boreal sedge-dominated
peatland. Foliar Hg concentrations decreased early in the growing season due to growth dilution. Foliar Hg concentrations were
subsequently positively correlated with leaf age (time). Hg concentrations were 1.4-1.7 times higher in sweet gale than in sedges.
A leaching experiment showed that sweet gale leached less Hg but more bioaccessible dissolved organic matter (DOM) by mass
than sedges. Leaching of Hg was positively related to the aromaticity of DOM in leachate, suggesting the importance of DOM
with higher aromaticity in controlling Hg mobility. Annual inputs of Hg through senesced leaf material to peat soils were 9.88
mg/ha/yr, 1.62 mg/ha/yr, and 8.29 mg/ha/yr for sweet gale, tussock sedge, and few-seeded sedge/wire sedge, respectively. Future
investigations into foliar Hg accumulation and input from other plant species to the sedge-dominated peatland are needed to
estimate the annual Hg inputs precisely.









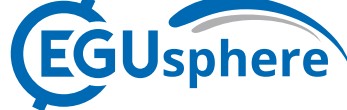

## 1 Introduction


Mercury (Hg), especially methylmercury (MeHg), is a global concern due to its potential toxicity and ubiquitous presence in the
environment (Morel et al., 1998). Hg is emitted to the atmosphere from both natural (e.g., volcanoes, wildfires, geothermal
activity) and anthropogenic sources (e.g., coal combustion, artisanal gold mining, incineration) (Schroeder and Munthe, 1998;
Streets et al., 2011). Atmospheric Hg exists as gaseous elemental mercury (GEM, Hg(0)), reactive gaseous mercury (RGM,
Hg(II)), and particulate-bound mercury (PBM, $Hg_p$) with GEM as the dominant species (> 95 %) (Schroeder and Munthe, 1998).
RGM and PBM have shorter atmospheric residence time ranging from hours to days, whereas GEM has a longer atmospheric
residence time of several months to a year and thus is transported globally (Schroeder and Munthe, 1998). These atmospheric Hg
species are eventually deposited into aquatic and terrestrial ecosystems via wet deposition (precipitation, such as rain, snow, and
fog) and dry deposition (particle settling or direct partitioning to vegetation, water, and soil surface, or direct absorption by
vegetation foliage) (Lindberg et al., 2007). Hg dry deposition is a larger input than wet deposition to vegetated terrestrial
landscapes, contributing 70 %~85 % of total Hg deposition (dry and wet deposition) in terrestrial ecosystems (Graydon et al.,
2008; Risch et al., 2017; Risch et al., 2012; St. Louis et al., 2001; Wang et al., 2016; Zhang et al., 2016), and more than 70 % of
Hg dry deposition is by vegetation litterfall/incorporation into soil organic matter (SOM) (Obrist et al., 2017; Wang et al., 2016).
Forest ecosystems are important sinks of atmospheric Hg and have received widespread attention from researchers (Risch et al.,
2012; St. Louis et al., 2001; Wang et al., 2016; Zhang et al., 2009); however, studies about foliar Hg accumulation in other plant
types or other ecosystems such as boreal peatlands are few (see Moore et al., 1995) despite their critical role in the carbon
(Gorham, 1991) and Hg cycles (Grigal, 2003). Boreal peatlands store 500 ± 100 Gt of carbon as peat (partially decomposed
vegetation matter) due to slow decomposition rates in their anaerobic and acidic conditions and low temperatures (Rydin and
Jeglum, 2013). In addition, boreal peatlands are sinks for inorganic Hg (St. Louis et al., 1994), and can be MeHg sources to
downstream ecosystems (Branfireun et al., 1996; Mitchell et al., 2008; St. Louis et al., 1994), given their anaerobic conditions,
non-limiting amounts of inorganic Hg, and often available but limited amounts of sulfate (Blodau et al., 2007; Schmalenberger et
al., 2007) and bioaccessible carbon (Mitchell et al., 2008).
Little information is available about the amount of atmospheric Hg accumulated in leaves in peatlands. Moore et al. (1995)
reported that Hg levels in nonvascular plants (fungi, lichens, and mosses) are almost an order of magnitude higher than those in
vascular plants in wetlands, and the Hg concentrations follow the sequence: grassland herbs < trees and shrubs < aquatic

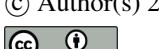



macrophytes < *Sphagnum* spp. mosses < lichens < fungi. However, the annual input of aboveground vegetation biomass to
peatland soils as senesced litter from vascular plants is greater than from bryophytes  (Frolking et al., 2001) in fen-type peatlands
dominated by sedges and shrubs, so despite having lower Hg concentrations, the mass input may be significant. With more
bioaccessible litter and leachate than bryophytes, vascular plant inputs may also decompose at faster rates, releasing Hg to the
soil and/or facilitating net methylation (Hobbie, 1996; Lyons and Lindo, 2019).
Boreal peatlands are experiencing rising temperatures due to climate change (IPCC, 2018) that is likely to either increase
aboveground biomass and leaf area in already vascular plant-dominated peatlands or potentially shift moss-dominated peatlands
to more vascular plant-dominated. Changes in plant abundance and community composition may further affect Hg deposition
(Zhang et al., 2016) and input through litterfall, given that foliar Hg concentrations (Moore et al., 1995) and annual vegetation
biomass input (Frolking et al., 2001) are different among plant species. Weltzin et al. (2000) found that the productivity of
vascular plants in peatlands increased under higher temperatures. A full factorial greenhouse laboratory experiment of increased
temperature and elevated atmospheric $CO_2$ resulted in increased peatland graminoid productivity (both above and belowground)
(Tian et al., 2020). Several studies have also shown that experimental warming of northern peatlands mesocosms altered plant
community composition, increasing vascular plant abundance and biomass, and decreasing *Sphagnum* spp. cover (Buttler et al.,
2015; Dieleman et al., 2015; Weltzin et al., 2000). Changes in plant abundance and biomass in northern peatlands are
ecologically significant, given their role as a source of net MeHg production.
Foliar Hg eventually enters peat soils via litterfall and is expected to follow the sequence: (1) wash-off of aerosols, particles, and
gases from leaf surfaces, (2) leaching of water-soluble components, and (3) incorporation into SOM after the microbial
decomposition of litter. Leaching is the initial phase of litter breakdown in aquatic environments and can rapidly release up to
30 % dissolved matter, primarily dissolved organic matter (DOM) within 24 h after immersion of litter (Gessner et al., 1999).
The amount of rapidly released Hg during litter leaching is unknown and needs to be elucidated because more recently deposited
Hg appears to be more readily methylated than "old" Hg in peat soils (Branfireun et al., 2005; Feng et al., 2014; Hintelmann et
al., 2002).
The overall objective of this study is to link the vascular plant community (i.e., sedges and shrubs) to the peatland Hg cycle in a
vascular plant-dominated fen-type peatland. We use "sedge-dominated fen" instead of "vascular plant-dominated fen-type
peatland" hereafter, given that sedges are the primarily dominant plants in this study site (Webster and McLaughlin, 2010). The
specific objectives of this study are to:



(1) quantify the mass accumulation of atmospherically-derived Hg in leaves of dominant plant species in a sedge-dominated fen
over a growing season;
(2) estimate the Hg input from the litter of different plant species and through litter leaching to peat soils;
(3) clarify the role of DOM characteristics in controlling Hg leaching;
(4) estimate the annual areal loading of foliar Hg of different plant species to peat soils.
**2 Materials and methods**
**2.1 Study site**
Samples were collected from a sedge-dominated fen (10.2 ha) located in an 817 ha sub-watershed of the Lake Superior basin
near White River Ontario, Canada (48°21' N, 85°21' W). The growing season is roughly from June to September. The sedge-
dominated fen is mostly open and the vegetation community is dominated by three sedge species: few-seeded sedge [*Carex*
*oligosperma* Michx.]; wire sedge [*Carex lasiocarpa* Ehrh]; and tussock sedge [*Carex stricta* Lamb.] (Lyons and Lindo, 2019).
Sweet gale [*Myrica gale* L.] is the dominant shrub at this site (Lyons and Lindo, 2019; Palozzi and Lindo, 2017). Details of the
study site and the characteristics of these plants are provided in the Supporting Information (SI). In this study, few-seeded sedges
and wire sedges were mixed during plant sample collection as they are indistinguishable in size and form from one another when
not in flower/seed, and frequently co-occur.
**2.2 Sample collection and analysis**
Five locations several hundred meters apart were selected in the sedge-dominated fen to serve as within-site replicates to account
for potential local-scale variability. These five locations were roughly evenly distributed over this study area. Approximately
fifty whole leaves of each few-seeded sedge/wire sedge, tussock sedge, and sweet gale were collected from each location using a
clean blade in the middle of June, July, August, and after senescence at the beginning of October 2018 in each location, totaling
60 samples. For the October sampling event, the sedge leaves were still standing with the lower sections green, and although
senesced, shrub leaves were sampled from the branch to ensure that there was no mixing with previous years' fallen leaves.
Disposable nitrile gloves were worn during the sample collection. All samples were double bagged with two polyethylene bags
and transported to the lab using a clean cooler. Leaves of each species that were collected from each plot in October 2018 were

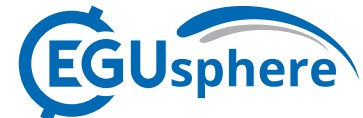

divided for foliar total Hg (THg) analyses and a foliar Hg leaching experiment. Leaves were stored frozen until they were
returned to the university laboratory.
For estimation of annual biomass of senesced leaf, seven 0.25 m$^2$ (0.5 m × 0.5 m) plots several hundred meters apart were
selected at the end of August 2019 during senescence and before leaf off. All aboveground biomass of few-seeded sedge/wire
sedge and tussock sedge and all aboveground leaf biomass of sweet gale were collected separately using a clean blade from each
0.25 m$^2$ plot. All vegetation samples were stored by species in paper bags, transported to the lab, and then oven-dried at 60 °C
for a minimum of 48 h. Dried leaves of each species in each plot were sorted and weighed to estimate senesced leaf biomass of
each species for each plot. The senesced leaf biomass of each species per hectare per year was calculated and expressed as
mg/ha/yr.
**Foliar total mercury, C content and N content.** In the laboratory, leaf samples for chemical analyses were rinsed three times
with deionized water (18.2 MΩ cm), freeze-dried for 48 h, ground and homogenized, and then analyzed using a Milestone™
DMA-80 (EPA method 7473). Leaf C content (%C; w/w) and N content (%N; w/w) before and after the foliar Hg leaching
experiment was analyzed using a CNSH analyzer (Vario Isotope Cube; Elementar). The ratio of leaf C content and N content
(C:N) were calculated. Detailed information concerning analytical methods are described in the SI, including analysis of foliar
total Hg, %C, and %N.
**Foliar mercury leaching experiment.** The foliar leaching experimental procedure followed the design of Rea et al. (2000) and
Del Giudice and Lindo (2017). Senesced leaves of sedges and sweet gale collected in October 2018 were rinsed twice with 100
mL of deionized water (18.2 MΩ cm) to quantify particulate or loosely-bound Hg and DOM that can be easily removed/leached
from the leaf surface. This water was reserved for subsequent analysis. After rinsing, the leaves were oven-dried at a low
temperature (40 °C) for 48 h, and then the leaves of each species from each location were relatively evenly separated into three
groups and weighed, totaling 45 groups. These oven-dried senesced leaf samples were immersed in 150 mL of deionized water
in clean 250 mL PETG bottles. All PETG bottles were capped, double bagged, and incubated in the dark at room temperature
(~21 °C) for 48 h. Senesced leaf materials were gently swirled at the beginning of the leaching experiment to ensure complete
wetting. Following the leaching, the leachate was vacuum filtered through a 0.45 μm glass fiber filter into clean 250 mL PETG
bottles. Leachate from each sample was split into two aliquots. One was preserved by acidifying to 0.5 % (vol/vol) with high-
purity HCl for dissolved total Hg (THg$_{aq}$) analysis and stored in 250 mL PETG bottles; the other was stored in the clean 60 mL
Amber glass bottles and analyzed within 2 d for the quantity and characteristics of DOM. All samples were stored in the dark at



4 °C for further analysis. Method blanks of the leaching experiment were performed at the same time following the same
procedure.
Senesced leaf material was taken out of each PETG bottle, oven-dried at 40 °C for 48 h, and re-weighed after leaching. The dry
leaf weight before and after the leaching process was used to calculate the mass loss. These re-dried senesced leaf samples after
leaching were ground and homogenized before the measurement for %C and %N as described above.
The dissolved total Hg (THg$_{aq}$) concentrations in the rinse water and leachate were analyzed using Environmental Protection
Agency (EPA) method 1631. Dissolved organic matter is quantified analytically as dissolved organic carbon (DOC). DOC
concentrations in rinse water and leachate were measured using an iTOC Aurora 1030 (OI Analytical, College Station, TX,
USA) using the persulfate wet oxidation method. Details on the analytical procedures and QA/QC data for concentrations of
THg$_{aq}$ and DOC are provided in the SI.
DOM in leachate was characterized as specific ultraviolet absorbance at a wavelength of 254 nm (SUVA$_{254}$), an indicator of the
molecular weight (or size) and aromaticity (the content of aromatic molecules) of DOM (Weishaar et al., 2003). Higher SUVA$_{254}$
values suggest that DOM contains more high-molecular-weight and aromatic molecules (Weishaar et al., 2003). Fluorescence
excitation-emission matrices (EEMs) were also collected for calculating informative optical indices that reflect differences in
DOM characteristics in leachate. The reported EEMs were then converted to optical indices using R Software (R Core Team
2012). Three common indices were chosen in this study: the fluorescence index (FI), the humification index (HIX$_{EM}$), and the
biological index or 'freshness' index (BIX). Lower FI values ($< 1.2$) indicate that DOM is terrestrially derived (resulting from
decomposition and leaching of plant and soil organic matter) and has higher aromaticity, while higher FI values ($> 1.8$) indicate
that DOM is microbially derived (originating from processes as extracellular release and leachate of algae and bacteria) and has
lower aromaticity (Fellman et al., 2010; McKnight et al., 2001). High HIX$_{EM}$ ($> 1.0$) values reflect the high humification of
DOM and DOM is composed of more highly condensed and higher molecular weight molecules (Fellman et al., 2010; Hansen et
al., 2016; Huguet et al., 2009; Ohno, 2002). Higher BIX  values ($> 1.0$) reflect that more low-molecular-weight DOM was
recently produced by microbes (Fellman et al., 2010; Huguet et al., 2009). Details on the analytical procedures and QA/QC data
for SUVA$_{254}$, FI, HIX$_{EM}$, and BIX are provided in the SI.



## 3 Statistical analysis

Results were analyzed using IBM SPSS statistics software (IBM SPSS Inc. 24.0). The repeated-measures ANOVA was performed to compare the difference in foliar THg concentrations among different plant species over the growing season and to analyze the effect of leaf age on foliar Hg concentrations. Linear regressions were analyzed to examine the relationship between foliar THg accumulation and leaf age. Differences in the foliage quality (%C, %N, and C:N) were analyzed using a multivariate ANOVA. One-way ANOVA was used to determine the effects of plant species on concentrations of $THg_{aq}$ and DOM quantity and characteristics in leachate. The repeated-measures ANOVA, multivariate ANOVA, and one-way ANOVA were followed by a *post hoc* test (Bonferroni's significant difference; honestly significant difference at the 95 % confidence interval). Weighed least squares regression was used to examine the nature of the relationship between $THg_{aq}$ concentrations and $SUVA_{254}$ in leachate. Data are presented as the mean ± standard deviation (SD). Coefficient of determination ($R^2$) and significance p-values (p) are presented for linear regression fits, and $p < 0.05$ was considered significant.

## 4 Results and discussion

### 4.1 Foliar mercury accumulation in peatland plants

Foliar THg concentrations were related to time/leaf age ($F_{(1.73,24.26)} = 42.75$, $p < 0.001$) and plant species ($F_{(1.23,23.38)} = 29.38$, $p < 0.001$) (Fig. 1). Based on *post hoc* tests, foliar THg concentrations were significantly different between plant species and between the sampling months, except that there was no significant difference in foliar THg concentrations between June and August. The mean foliar THg concentrations (n = 5) in June followed the sequence: few-seeded sedge/wire sedge < tussock sedge < sweet gale. In July foliar THg concentrations decreased by 30 % (few-seeded sedge/wire sedge), 40 % (tussock sedge), and 47 % (sweet gale), respectively. The decrease in THg concentrations is likely because of leaf growth dilution, although changes in leaf biomass were not quantified as part of this study. Foliar THg concentrations were positively related to time after July (few-seeded sedge/wire sedge: $F_{(1,13)} = 185.79$, $p < 0.001$, $R^2 = 0.93$; tussock sedge: $F_{(1,13)} = 200.87$, $p < 0.001$, $R^2=0.94$; sweet gale: $F_{(1,13)} = 70.72$, $p < 0.001$, $R^2=0.84$). The mean foliar THg concentrations in October few-seeded sedge/wire sedge, tussock sedge, and sweet gale were 1.7, 1.3, and 2.0 times higher than the initial concentrations in June. This result showed a clear pattern of continuous THg accumulation from the atmosphere over time as has been shown for forests (Laacouri et al., 2013; Millhollen et al., 2006b; Rea et al., 2002), given that plant roots act as a barrier of Hg transport from soils to shoots (Wang et al., 2015).



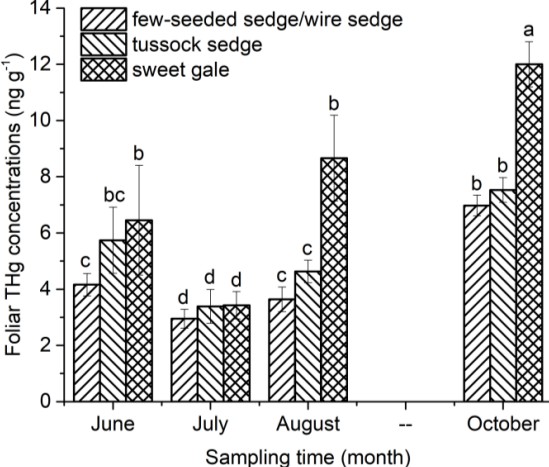


**Figure 1 The intraseasonal trend in foliar total mercury (THg) concentrations (ng g⁻¹) of few-seeded sedge/wire sedge, tussock sedge,**

**and sweet gale (ng g⁻¹). All concentrations are expressed in dry weight. Error bars are ± SD (n = 5 for each species for each time**

**interval). The same letters above bars denote that values of foliar THg concentrations are not significantly different at the 0.05 levels.**

Mercury accumulation in leaves is affected by many factors, such as atmospheric Hg concentration, environmental conditions
(e.g., solar radiation and temperature), and biological factors (e.g., leaf age, plant species, leaf area, and leaf placement)
(Blackwell and Driscoll, 2015; Ericksen et al., 2003; Ericksen and Gustin, 2004; Laacouri et al., 2013; Millhollen et al., 2006a).
Since all samples were collected in the same location, factors such as atmospheric Hg concentration and environmental
conditions were deemed the same, leaving only biological factors as an explanation for differences.
**Leaf age.** Leaf age is an important biological factor in controlling foliar concentrations (Ericksen et al., 2003; Laacouri et al.,
2013). The positive relationship between foliar THg concentrations and time after July suggests that leaves of all species here
continued to assimilate atmospheric Hg over the growing season right up to senescence. Some studies have found that the rate of
foliar Hg uptake decreased toward the end of the growing season (Ericksen et al., 2003; Laacouri et al., 2013; Poissant et al.,
2008), which appears to be because of the decrease of photosynthetic activity at the end of the growing season (Koike et al.,
2003). Despite the decline of foliar Hg uptake during the late growing season, foliar Hg concentrations continue to increase right
up to senescence because of the immobilization of the majority of foliar Hg (Laacouri et al., 2013; Lodenius et al., 2003;
Stamenkovic and Gustin, 2009).
**Plant species.** Plant photosynthesis, transpiration, growth rates, and leaf area are different among plant species (Antúnez et al.,
2001; Laacouri et al., 2013; Millhollen et al., 2006b), and given that these are important controls on Hg accumulation, the





differences among species found in this study are not surprising. The mean foliar THg concentrations in tussock sedge were 1.2
times higher than that in few-seeded sedge/wire sedge, and although not measured as part of this study, tussock sedge has a
larger leaf area than few-seeded sedge/wire sedge (Newmaster et al., 1997). A larger leaf has more stomates and thus more leaf
accumulation of atmospheric Hg (Laacouri et al., 2013; Millhollen et al., 2006; Stamenkovic and Gustin, 2009). The higher
relative Hg concentrations in sweet gale (mean 1.7 and 1.4 times higher than few seeded/wire, and tussock sedge, respectively) is
likely due in part to the same leaf area relationship. In addition, Kozlowski and Pallardy (1997) reported that leaves near the top
of the canopy generally have higher rates of photosynthesis and stomatal conductance than those near the bottom of the canopy
due to light saturation. Sweet gale had potentially higher stomatal conductance due to higher incident radiation and vapor
pressure deficits than sedges that are lower to the saturated ground with tightly packed vertical leaves.
Concentrations of Hg in senesced leaves of few-seeded sedge/wire sedge, tussock sedge, and sweet gale (6.58 ng g$^{-1}$ to 12.77 ng
g$^{-1}$) were lower than that reported in tree litter (17 ng g$^{-1}$ – 238 ng g$^{-1}$) (Laacouri et al., 2013; Obrist et al., 2021; Poissant et al.,
2008; Rea et al., 2002; Wang et al., 2016; Zhang et al., 2009) but similar to that previously reported for sedges and shrubs in
Canada (10.2 ± 6.8 ng g$^{-1}$) (Moore et al., 1995). The foliar Hg concentrations for plant species in this study increased 1.3-2.0
times over the growing season, which was smaller than that (3-11 fold) reported for trees (Laacouri et al., 2013; Poissant et al.,
2008; Rea et al., 2002). The above results suggested that foliar Hg concentrations differ among vegetation types (Demers et al.,
2007; Moore et al., 1995; Obrist et al., 2012; Richardson and Friedland, 2015), which might be attributed to the larger leaf and
higher stomatal density/ leaf placement in trees than sedges and shrubs.
**Leaf carbon, nitrogen and mercury.** Leaf %C, %N, and C:N were significantly different among plant species ($F_{(6,104)}$ = 59.64, p
< 0.001) over the growing season ($F_{(9,124)}$ = 45.42, p < 0.001) (Fig. 2). Based on *post hoc* tests, foliar %C, %N, and C:N was
significantly different between sweet gale and sedges (few-seeded sedge/wire sedge and tussock sedge) but not between few-
seeded sedge/wire sedge and tussock sedge. Foliar %C and %N were much lower in these sedges than sweet gale, which agrees
well with a previous study that deciduous shrubs (i.e., sweet gale) generally have a higher foliar %C and %N than grasses
(Wright et al., 2004). The fixation of nitrogen in sweet gale is in part attributed to sweet gale root nodules containing symbiotic
nitrogen-fixing (Newmaster et al., 1997; Vitousek et al., 2002) with this greater amount of available N leading to higher
photosynthetic capacity (Wright et al., 2004), thus, species containing a higher foliar %N are usually accompanied with a
higher %C.





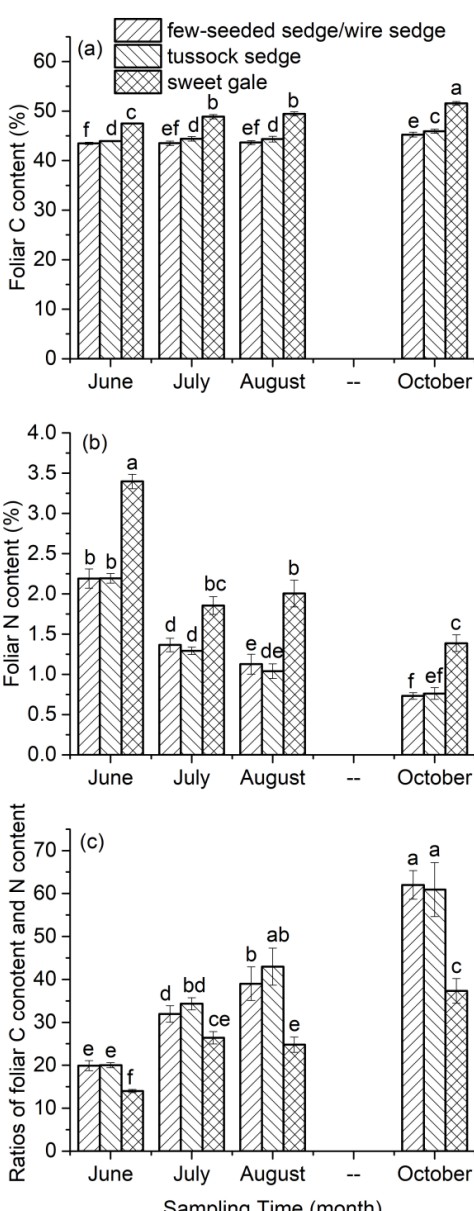


**Figure 2 The carbon content (%C) (a), nitrogen content (%N) (b), and the ratio of carbon content to nitrogen content (C:N) (c) over the 2018 growing season. Vertical bars are mean ± SD (n = 5). The same letters above bars denote that values of foliar THg concentrations are not significantly different at the 0.05 levels.**

There were significant increases in foliar %C (few-seeded sedge/wire sedge: $F_{(3,9)}$ = 25.98, p < 0.001; tussock sedge: $F_{(3,9)}$ = 20.56, p < 0.001; sweet gale: $F_{(3,9)}$ = 115.90, p < 0.001) but sharp decreases of foliar %N (few-seeded sedge/wire sedge: $F_{(1.34,4.03)}$



=354.20, p < 0.001; tussock sedge: $F_{(3,9)}$ = 252.36, p < 0.001; sweet gale: $F_{(3,9)}$ = 170.43, p < 0.001) over the growing season (Fig.
2). The strong decreases in foliar %N with leaf age can be attributed to the translocation of N from senescing leaves to new
leaves (Wang et al., 2003). A study found that approximately 77 % N, 57 % phosphorus (P), and 44 % potassium (K) were
translocated out of senescing leaves during mangrove leaf senescence (Wang et al., 2003). Foliar C is sequestrating continuously
over the growing season (Kueh et al., 2013). The element re-translocation and C sequestration in leaves may lead to the
foliar %C increase with time. The values of foliar C:N increased with time, which is a function of the decreases of foliar %N and
the increases of foliar %C.
Senesced leaf tissue with higher foliar %C and %N had higher foliar THg concentrations (%C and Hg: $F_{(1,13)}$ = 191.09, p < 0.05,
y = 0.78x − 28.20, $R^2$ = 0.94;%N and Hg: $F_{(1,13)}$ = 82.38, p < 0.05, y = 7.16x – 1.96, $R^2$ = 0.93) (Fig. 3a and 3b). THg
concentrations were negatively related to foliar C:N during senescence ($F_{(1,13)}$ = 175.10, p < 0.05, y = 0.18x – 18.33, $R^2$ = 0.86;
Fig. 3c). A previous study found soil Hg concentrations were positively related to soil organic C and N, and then given a possible
explanation that high C and N levels in soil reflect high vegetation productivity corresponding with high atmospheric Hg
deposition via litterfall (Obrist et al., 2009). Although the mechanism of these relationships between Hg concentrations and
contents of C and N in senesced leaves materials is still unclear, this study shows that higher C and N content in senesced leaves
indirectly indicates a higher input of Hg via litterfall to soils.





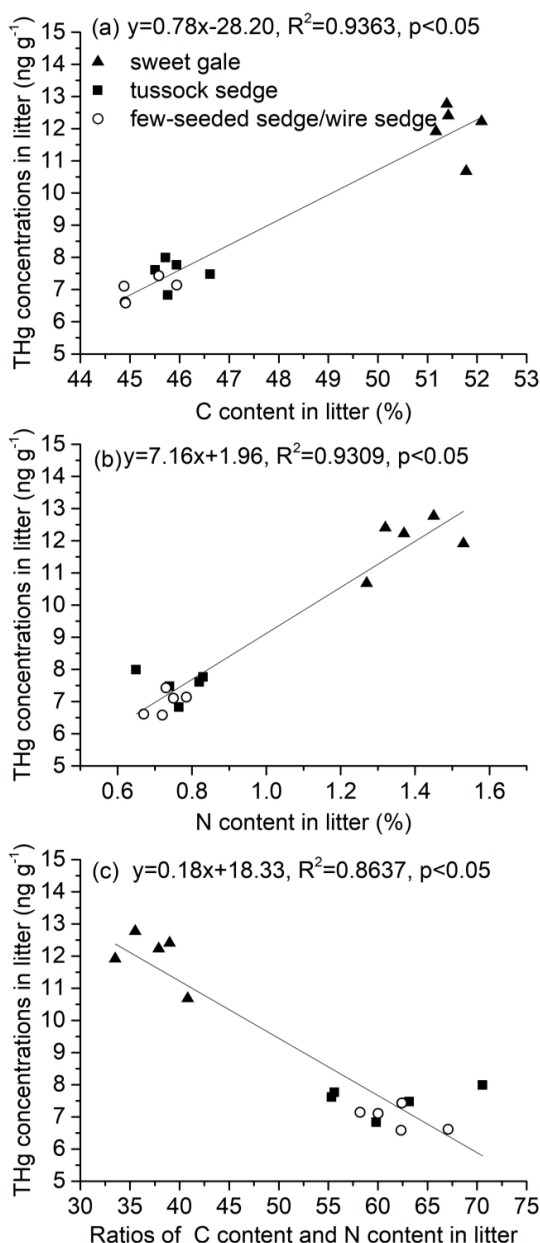


**Figure 3 Correlations between (a) THg concentrations and C contents, (b) THg concentrations and N contents, and (c) THg**

**concentrations and ratios of C content and N content (C:N) in litter. All linear correlations are statistically significant (p < 0.05).**

**4.2 Mercury leaching from senesced leaves**

**Surface-rinsable mercury.** The mean mass of Hg from the surface rinse of senesced leaf material (expressed per gram of dry

senesced leaf) was $0.02 \pm 0.01$ ng g$^{-1}$ and $0.01 \pm 0.00$ ng g$^{-1}$ (or $3.27 \pm 1.68$ ng L$^{-1}$ and $1.39 \pm 0.83$ ng L$^{-1}$, expressed per liter of



rinse water (18.2 MΩ cm)), respectively, indicating that mass of Hg that was loosely bound on the leaf surface was small relative
to the total senesced leaf tissue Hg concentration (8.83 ± 2.38 ng g$^{-1}$) representing on average only 0.4 % Hg (tussock sedge:
0.6 %; few-seeded sedge/wire sedge: 0.3 %; sweet gale: 0.3 %) of total THg mass.
**Leachable mercury.** The mean THg$_{aq}$ mass per gram of senesced leaf had significant differences between plant species (F$_{(2,41)}$ =
11.55, p < 0.001; Fig. 4). Based on *post hoc* tests, there were significant differences in THg$_{aq}$ mass per gram of senesced leaf
between sweet gale and sedges (few-seeded sedge/wire sedge and tussock sedge) but not between few-seeded sedge/wire sedge
and tussock sedge. The senesced leaf of sweet gale leached the least Hg among these plant species, which is likely due to their
hydrophobic waxy cuticle that may both retain Hg, as well as protect the inner leaf material from leaching. Another plausible
explanation is that N was more easily released from sedges than C and it was the opposite for sweet gale, based on changes in
foliar %C and %N between before and after leaching (Table 1), whereas N groups in litter hinder the leaching of foliar Hg
(Obrist et al., 2009). Foliar %N of sweet gale increased after leaching, which is likely attributed to a large amount of loss of other
elements, such as K, Mg, and P, although they were not part of this experiment. Bessaad and Korboulewsky (2020) found that
60–79 % of K, 19–50 % of Mg, 22–30 % of P, and < 16 % of Ca and N were leached out from fully developed broadleaves
(collected in summer) during rainfall.

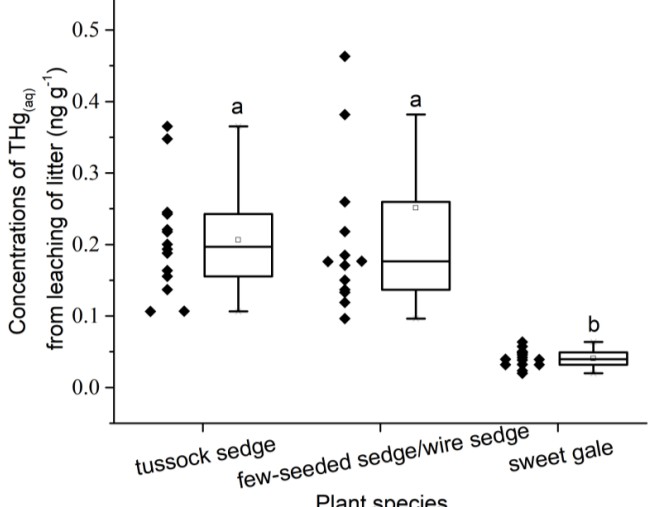


**Figure 4 Mass of mercury leached per gram of senesced leaf material (ng g$^{-1}$). Boxplot displays median (50th percentile; the inside line**
**of the box), first quartile (25th percentile; lower bound of the box), third quartile (75th percentile; upper bound of the box), whiskers**
**(all measures between 5th percentile and 25th percentile and between 75th percentile and 95th percentile; the straight line below and**
**above the box), and outliers (individual points outside of the percentile of 5th and 95th). n = 15.**





**Table 1 Changes of foliar carbon content (%C) and nitrogen content (%N) during leaching of litterfall. n = 15**

| | foliar %C | | foliar %N | |
|---|---|---|---|---|
| | before leaching | after leaching | before leaching | after leaching |
| sweet gale | 51.57 ± 0.36 | 51.03 ± 0.34 | 1.39 ± 0.10 | 1.50 ± 0.07 |
| tussock sedge | 45.91 ± 0.42 | 44.97 ± 0.54 | 0.76 ± 0.07 | 0.68 ± 0.10 |
| few-seeded sedge/wire sedge | 45.24 ± 0.49 | 43.83 ± 0.49 | 0.73 ± 0.04 | 0.64 ± 0.02 |


During experimental leaching, 3.0 %, 2.9 %, and 0.3 % of the total THg mass present in tussock sedge, few-seeded sedge/wire
sedge, and sweet gale senesced leaf was leached, respectively. The percentages of Hg that leached from tussock sedge, few-
seeded sedge/wire sedge leaves were 5.5 and 10.6 times higher than that from rinses, while the percentage of Hg that leached
from sweet gale senesced leaf was similar to that from rinse water (0.3 %). Rea et al. (2000) reported that surface washoff of
loosely bound and particulate Hg was a rapid and larger source of Hg in forest throughfall compared to continuously foliar Hg
leaching from live leaves. It is likely because dry leaves lack structural integrity compared to live leaves in Rea et al.'s (2000)
study, leading to more rapid leaching of soluble constituents (Gessner et al., 1999), including Hg, so the results of these prior
studies are not directly comparable to this one. Further, although Hg leached from senesced leaf material was a small (< 5 % of
foliar tissue Hg) but a measurable contributor to the mass balance, it is one that would be completely missed if material had been
collected from a litter trap that had been exposed to rainfall for any period.
**4.3 Quantity and characteristics of leachate dissolved organic matter**
**The quantity and characteristics of DOM in leachate.** The mean mass of DOC leached per gram of senesced leaf material and
the mass loss during senesced leaf material leaching was significantly different between plant species (leached DOC mass: $F_{(2,42)}$
= 34.95, p < 0.001; mass loss: $F_{(2,42)}$ = 11.62, p < 0.05) with a same sequence following: few-seeded sedge/wire sedge < tussock
sedge < sweet gale (Fig. 5). The same sequence is in part because the loss of soluble carbons accounted for the majority of the
mass loss during litter leaching (Del Giudice and Lindo, 2017). Mass loss of sweet gale (17.7%) was significantly larger than
sedges (few-seeded sedge/wire sedge (8.1%) and tussock sedge (11.5%)). The released DOC accounted for 22.96 ± 14.85%,
23.73 ± 12.95%, and 17.03 ± 6.68% of mass loss during senesced leaf material leaching for few-seeded sedge/wire sedge, tussock
sedge, and sweet gale, respectively. Loss of other nutrients, such as dissolved organic nitrogen (DON) and dissolved organic




phosphorus (DOP) (Ong et al., 2017; Liu et al., 2018; Hensgens et al., 2020) and the inorganic components and other elemental
organic matter (Lavery et al., 2013; Jiménez et al., 2017) also contribute to the mass loss, despite these nutrients not being
measured.

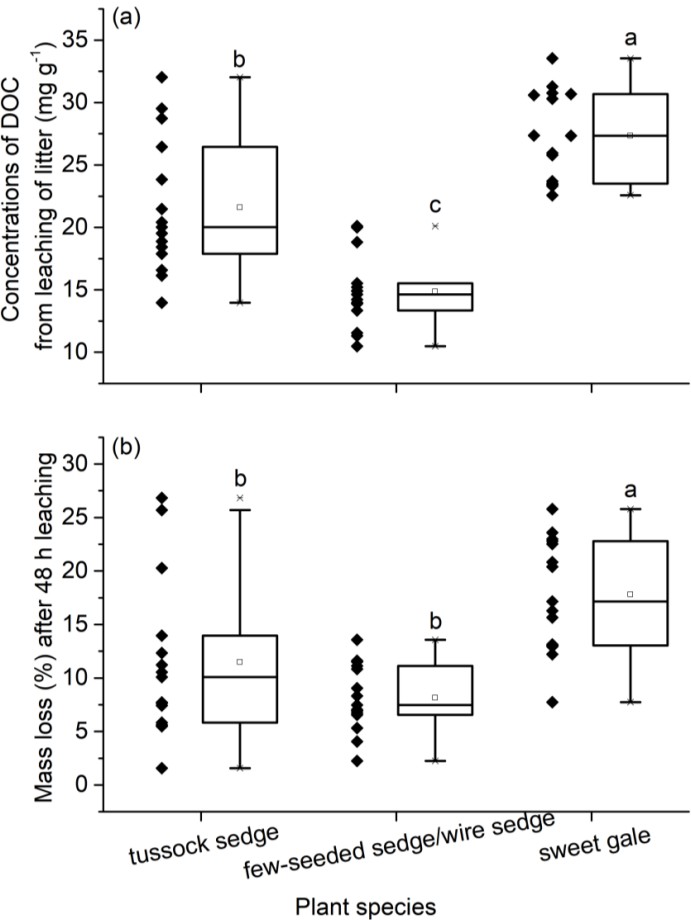


**Figure 5 Mass of dissolved organic carbon (DOC) leached per gram of senesced leaf material (mg g⁻¹) (a), and mass loss after 48 h**
**leaching (b). Boxplot displays median (50th percentile; the inside line of the box), first quartile (25th percentile; lower bound of the**
**box), third quartile (75th percentile; upper bound of the box), whiskers (all measures between 5th percentile and 25th percentile and**
**between 75th percentile and 95th percentile; the straight line below and above the box), and outliers (individual points outside of the**
**percentile of 5th and 95th). n = 15.**
Characteristics of DOM also varied among plant species (SUVA$_{254}$: $F_{(2,42)}$ = 24.02, p < 0.001; HIX$_{EM}$:$F_{(2,42)}$ = 3.82, p < 0.05; FI:
$F_{(2,42)}$ = 11.24, p < 0.001; BIX: $F_{(2,42)}$ = 125.48, p < 0.001) (Fig. 6 and Table 2). Based on *post hoc* tests, there were significant



differences in SUVA$_{254}$ between sweet gale and sedges (few-seeded sedge/wire sedge) only and in BIX among all plant species;
there were no significant differences in HIX$_{EM}$ among plant species.

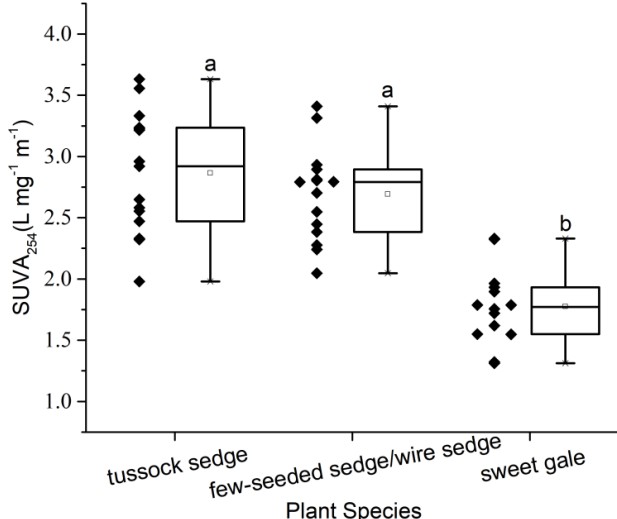


**Figure 6 Dissolved organic matter characteristics as measured by specific ultraviolet absorbance at the wavelength 254 nm (SUVA$_{254}$),**
**n = 15. Boxplot displays median (50th percentile; the inside line of the box), first quartile (25th percentile; lower bound of the box),**
**third quartile (75th percentile; upper bound of the box), whiskers (all measures between 5th percentile and 25th percentile and**
**between 75th percentile and 95th percentile; the straight line below and above the box), and outliers (individual points outside of the**
**percentile of 5th and 95th).**
**Table 2 The mean fluorescence indices of dissolved organic matter characteristics[a]**

| Index | Tussock sedge | Few-seeded sedge/wire sedge | Sweet gale |
|---|---|---|---|
| FI | 1.19 ± 0.10 | 1.31 ± 0.09 | 1.49 ± 0.27 |
| HIX$_{EM}$ | 0.16 ± 0.03 | 0.16 ± 0.02 | 0.19 ± 0.03 |
| BIX | 0.53 ± 0.05 | 0.63 ± 0.06 | 0.35 ± 0.04 |

[a]Lower values of the FI (< 1.2) suggest dissolved organic matter (DOM) has higher aromaticity and is primarily composed of
high-molecular-weight DOM, while high FI values (> 1.8) indicate that DOM has lower aromaticity and is mainly composed of
low-molecular-weight DOM. DOM with high HIX$_{EM}$ (> 1) values is composed of more highly condensed and higher molecular
weight molecules. In contrast, higher BIX (> 1.0) values reflect that more low-molecular-weight DOM is recently produced,
generally, by microbes. All indices are unitless, n = 15.



The mean value of SUVA$_{254}$ in leachate followed the sequence: tussock sedge > few-seeded sedge/wire sedge > sweet gale
leaves, respectively, indicating that leached DOM from tussock sedge and few-seeded sedge/wire sedge leaves had higher
aromaticity and less bioaccessible than that from the sweet gale leaves. These results are supported by indexes of FI and HIX$_{EM}$.
DOM in senesced leaf material leachate of tussock sedge and few-seeded sedge/wire sedge had lower values of FI and HIX$_{EM}$
than that of sweet gale leaves, indicative of the presence of less bioaccessible and more aromatic DOM contents in sedges than in
sweet gale. All BIX values (0.26–0.73) measured in this study were lower than 1.0, reflecting that DOM is mainly terrestrially
derived (leaching from litterfall) in this study. Although DOM leached from different litters has different characteristics, DOM
leaching from litters is a substantial source to surrounding ecosystems (Davis et al., 2003; Davis et al., 2006; Del Giudice and
Lindo, 2017). Importantly, the leached DOM (e.g., organic acids, sugars; amino acids) can provide energy and nutrients for
microbes (Davis et al., 2003), which will subsequently stimulate biological degradation and Hg methylation.

**4.4   Correlation between THg$_{aq}$ concentrations and SUVA$_{254}$ values in leachate.**

The concentrations of soluble THg$_{aq}$ were significantly related to SUVA$_{254}$ values ($F_{(1,41)}$ = 52.06, p < 0.001, y = 0.09x – 0.09, $R^2$
= 0.55; Fig. 7). Hg is tightly and readily bound to reduced sulfur groups (i.e., thiols) in DOM (Ravichandran, 2004; Xia et al.,
1999), especially those with higher aromaticity that have more reduced sulfur groups (Dittman et al., 2009). Mercury weakly
binds to carboxyl and phenol functional groups in DOM after all thiol groups are occupied at relatively high Hg concentrations
(Drexel et al., 2002; Graham et al., 2012), which is atypical in most natural environments in which Hg concentrations are
relatively low. This result agreed well with the literature indicating that DOM with higher aromaticity plays an important role in
controlling Hg mobility, given that the number of reduced sulfur groups far exceeds the amount of Hg in natural environments
(Ravichandran, 2004).





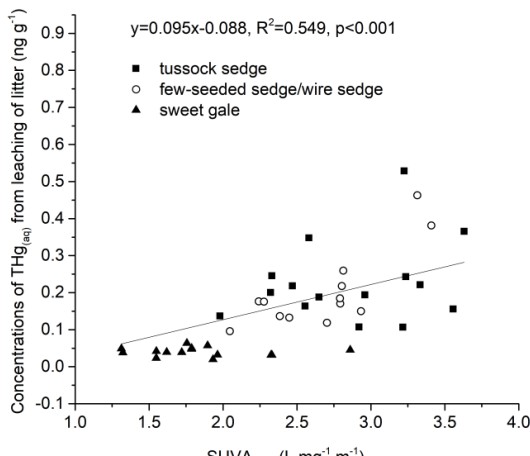


**Figure 7 Correlations between the mass of mercury leached per gram of senesced leaf material (THg$_{aq}$) and the specific ultraviolet**

**absorbance at the wavelength 254 nm (SUVA$_{254}$) in leachate.**

**4.5 Estimation of annual input of Hg via senesced leaves and rapid Leaching to peat soils**

The annual input of leaf biomass (mg/ha/yr) of few-seeded sedge/wire sedge into peat soils was 5.55 and 1.41 fold higher than

tussock sedge and sweet gale, while the annual inputs of Hg (mg/ha/yr) via sweet gale leaves were 6.29 and 1.22 fold higher than

via tussock sedge and few-seeded/wire sedge leaves in the sedge-dominated fen (Table 3). Annual total Hg input through

senesced leaves to peat soils were 1.56, 8.03, and 9.82 mg/ha/yr for tussock sedge, few-seeded sedge/wire sedge, and sweet gale,

respectively. The input of surficial Hg and leachable Hg accounted for 0.64 % and 0.37 %, 0.31 % and 3.20 %, and 2.86 % and

0.30 % of total foliar Hg input to peat soils annually for tussock sedge, few-seeded sedge/wire sedge, and sweet gale,

respectively. The majority of Hg in senesced leaves (> 96 %) was from the deposition of solid plant tissues in litter.







**Table 3 Annual input of senesced leaves, and senesced leaf Hg, surficial Hg, and leached Hg during leaching into peat**
**soils per hectare and per year in the sedge-dominated fen (mg/ha/yr).**

| Species | Senesced leaf biomass (mg/ha/yr) | Litter total Hg input (mg/ha/yr) | Washoff Hg input (mg/ha/yr) | Leachate Hg input (mg/ha/yr) |
|---|---|---|---|---|
| Tussock sedge | $2.07 \times 10^8$ | 1.56 | 0.01 | 0.05 |
| Few-seeded sedge/wire sedge | $1.15 \times 10^9$ | 8.03 | 0.03 | 0.23 |
| Sweet gale | $8.18 \times 10^8$ | 9.82 | 0.03 | 0.03 |
| Total | $2.17 \times 10^9$ | 19.41 | 0.07 | 0.31 |


Based on the data from the study growing season, the annual input of Hg in total via senesced leaves (19.40 mg/ha/yr) was 5-
22 % of litterfall in forest ecosystems (e.g., jack pine/black spruce/balsam fir forest, red maple/birch forest, Norway spruce
forest; 86-372 mg/ha/yr) (St Louis et al., 2001; Graydon et al.,2008; Shanley and Bishop, 2012), which can be attributed to those
forest ecosystems having both higher mean foliar Hg concentrations (21-51 ng g$^{-1}$) (Zhou and Obrist, 2021) and much greater
aboveground biomass and litterfall inputs (2000-3488 kg/ha/yr) (Graydon et al., 2008) than plants in this study. The overall
annual Hg inputs via these senesced leaves to peat soils in this sedge-dominated fen were 59 % of that via wet deposition using
the mean precipitation Hg input estimates from the Experimental Lakes Area (33 mg/ha/yr) for the years 2001-2010, which is in
the same general geographic region of Ontario (St Louis et al., 2019).
**5 Conclusions**
This study shows that the widely-observed pattern of foliage accumulation of Hg from the atmosphere and changes in foliar Hg
concentrations over time are the same in peatland vascular plants as they are for forest trees and that the patterns are related to
time/leaf age and plant species. Although THg concentrations in litterfall in this study are relatively lower than that in the forest
litterfall, Hg input through litterfall to peatland soils cannot be neglected, given that peatlands are "hotspots" of MeHg
production (Mitchell et al., 2008). Foliar leaching of lower molecular weight DOM from peatland shrubs such as sweet gale
provides energy for bacteria (including Hg methylators) and can enhance microbial metabolism. Hg released from ubiquitous



sedge litter during leaching is relatively more quickly than the much slower release of tissue-associated Hg through the
decomposition of plant tissues. Thus, the supply of inorganic Hg to sites of methylation in peatlands has both fast and slow
pathways that may shift under climate change, given that peatland plant species composition and biomass will certainly change
under climate change.

■DATA AVAILABILITY
All data generated or analysed during this study are included in this published article and its supplementary information files.
■SUPPLEMENT
The supplement related to this article is available online.
■AUTHOR CONTRIBUTION
Ting Sun carried the project out, collected all samples, performed the leaching experiment, analyzed samples and data, and wrote
the manuscript. Brian A. Branfireun designed the experiments, provided supervision, and edited the manuscript.
■COMPETING INTERESTS
The authors declare that they have no conflict of interest.
■DISCLAIMER
Publisher's note: Copernicus Publications remains neutral with regard to jurisdictional claims in published maps and institutional
affiliations.
■ACKNOWLEDGMENTS
The research is supported by the Natural Sciences and Engineering Research Council of Canada (NSERC) Strategic projects
Grant (STPGP/479026-2015). We thank all members of Dr. Brian A. Branfireun and Dr. Zoë Lindo lab group for their help in
the field.



■ABBREVIATIONS
Hg, mercury; MeHg, methylmercury; GEM, gaseous elemental mercury; RGM, reactive gaseous mercury; PBM, particulate-
bound mercury; THg, total mercury; $THg_{aq}$, dissolved total mercury; SRB, sulfate-reducing bacteria; %C, carbon content; %N,
nitrogen content; C:N, the ratio of leaf C content and N content; dissolved organic matter (DOM); DOC, dissolved organic
carbon; $SUVA_{254}$, specific ultraviolet absorbance at a wavelength of 254 nm; EEMs, fluorescence excitation-emission matrices;
FI, fluorescence index; $HIX_{EM}$, humification index; BIX, biological index; soil organic matter (SOM); CRM, certified reference
material; RSD, relative standard deviation.















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
