# Peer review of "Plant mercury accumulation and litter input to a Northern Sedge-dominated Peatland"

_EGUsphere, 2022_

## Author Comment (AC2)

Reviewer 2

We are grateful for the reviewer's deeply informed and insightful comments. We have addressed the detailed comments that were made on the PDF copy of the earlier version of the manuscript. Please find our replies to your comments and suggestions below.

Line 23, you can change "." to ",".

*Reply: We agree with this suggestion.The clarified text (Page 2, Line23) reads as follows "Plant foliage plays an essential role in accumulating mercury (Hg) from the atmosphere and transferring it to soils in terrestrial ecosystems, while many studies have focused on forested ecosystems.".*

Line 34: where these Hg accumulated in plants come from?

*Reply: Thank you for the reviewer's question. Vegetation is generally considered a sink for atmospheric Hg, with the majority of Hg in vegetation leaves accumulated from the atmosphere (Jiskra et al., 2018; Obrist et al., 2017). Previous studies showed that plant roots acted as a barrier of Hg transport from soils to shoots (Wang et al., 2015), and less than 10% of Hg in roots was transported to the aboveground portion of plants (Ericksen et al., 2003; Mao et al., 2013).We have incorporated additional content in Page 3 (Line 56-65). The clarified text (Page 3, Line 56-65) reads as follows "Vegetation is generally considered a sink for atmospheric Hg, with the majority of Hg in vegetation leaves accumulated from the atmosphere (Jiskra et al., 2018; Obrist et al., 2017). Plant leaves accumulate Hg from the atmosphere mainly through stomatal uptake (Lindberg et al., 1992). Stamenkovic and Gustin (2009) suggested that the non-stomatal pathway of Hg deposition to the leaf cuticle and subsequently retention and incorporation into leaf tissue also plays an important role in accumulating atmospheric Hg. Plant roots generally acted as a barrier of Hg transport from soils to shoots (Wang et al., 2015), and less than 10% of Hg in roots was transported to the aboveground portion of plants (Ericksen et al., 2003; Mao et al., 2013). Some studies also found that a great proportion of foliar Hg in Halophytes in salt marshes was translocated from the root (Canário et al., 2017; Cabrita et al., 2019; Weis and Weis 2004). The plausible reason is that plants in the hydroponic growth system have fewer apoplastic barriers (i.e. Casparian bands and suberin lamellae) in root architecture than plants grown in contaminated soils (Redjala et al., 2011)".*

Line 65 Here are more Hg accumulated studies in the peatland"

Grigal, D., Kolka, R., Fleck, J. & Nater, E. Mercury budget of an upland- peatland watershed. Biogeochemistry 50, 95–109 (2000).

Grigal, D. F. Mercury sequestration in forests and peatlands: A review. J. Environ. Qual. 32, 393–405 (2003)

Osterwalder, S. et al. Mercury evasion from a boreal peatland shortens the timeline for recovery from legacy pollution. Sci. Rep. 7, 16022 (2017).

Woerndle, G. E. et al. New insights on ecosystem mercury cycling revealed by stable isotopes of mercury in water flowing from a headwater peatland catchment. Environ. Sci. Technol. 52, 1854–1861 (2018).

*Reply:We agree with this suggestion. We have incorporated additional content in this section and cited more recent references. The clarified text (Page 4, Line 75-77) reads as follows "Previous studies have found that the majority of Hg in plant leaves in wetlands was from the atmosphere (Brahmstedt et al., 2021; Enrico et al., 2016; Fay and Gustin 2007) and nonvascular plants (e.g., fungi, lichens, and mosses) had higher foliar Hg concentrations than vascular plants (Moore et al., 1995; Pech et al., 2022).".*

Line 118. Plot size sampling in 2018? Is that the same size with samples collected in 2019?

*Reply:Thanks for the reviewer's question. Actually, the "plot" in 2018 should be "location". In 2018, five locations several hundred meters apart were selected in the sedge-dominated fen to serve as within-site replicates to account for potential local-scale variability. In 2019, seven 0.25 m2 (0.5 m × 0.5 m) plots several hundred meters apart were selected at the end of August 2019 during senescence and before leaf off to estimate the annual biomass of senesced leaf. The revised text (Page 6, Line 129) reads as follows "Leaves of each species that were collected from each location in October 2018 were divided for foliar total Hg (THg) analyses and a foliar Hg leaching experiment.".*

Line 121. For annual biomass, there is always variation between different years, depends on weather conditions, such wet or drought, warm or cool, etc.

*Reply:We agree with this suggestion. However, based on the weather data, the mean air temperature in the growing season in 2018 and 2019 was 15.82 ± 3.50 °C and 15.16 ± 3.38 °C, respectively. The total precipitation in the growing season in 2018 and 2019 was 243.9 mm and 189.3 mm, respectively. The mean water table levels in the growing season in 2018 and 2019 were –6.3 ± 1.0 cm, and –7.5 ± 3.9 cm, respectively. There were no significant differences in air temperature and precipitation between the growing season in 2018 and 2019 (temperature: $F_{(1,182)}$ = 1.74, p > 0.05; precipitation: $F_{(1,182)}$ = 0.48, p > 0.05). Therefore, we think the annual plant biomass may not change significantly between 2018 and 2019.*

*We have incorporated additional content in the Supporting Information (Page 1, Line 7-13) and the clarified text reads as follows "The mean annual air temperature and precipitation from 2012 to 2018 were 1.7 °C and 721 mm, respectively. The mean air temperature in the growing season in 2018 and 2019 was 15.82 ± 3.50 °C and 15.16 ± 3.38 °C, respectively. The total precipitation in the growing season in 2018 and 2019 was 243.9 mm and 189.3 mm, respectively. The mean water table levels in the growing season in 2018 and 2019 were –6.3 ± 1.0 cm, and –7.5 ± 3.9 cm, respectively. There were no significant differences in air temperature and precipitation between*

*the growing season in 2018 and 2019 (temperature: $F_{(1,182)} = 1.74$, $p > 0.05$; precipitation: $F_{(1,182)} = 0.48$, $p > 0.05$).".*

Line 183. Meaning of F(1.73,24.26)and F(1.23,23.38)?

*Reply:Thank you for the reviewer's question. For the repeated-measures ANOVA, we get the results F(x,y) and p. F is the statistics of the repeated-measures ANOVA, p-value indicates that if the mean difference among all groups was statistically significant. x and y is calculated based on the degree of freedom between subjects and the degree of freedom within subjects. Based on Mauchly's Test of Sphericity, if the results did not follow Mauchly's Test of Sphericity, the results needed to be modified with epsilon (e). We double-checked our analysis throughout the manuscript and made the necessary corrections in the manuscript. The revised text (Page 8, Line 192) reads as follows "Foliar THg concentrations were related to time/leaf age ($F_{(3,36)} = 108.86$, $p < 0.001$) and plant species ($F_{(2,12)} = 51.85$, $p < 0.001$) (Fig. 1)".*

Line 189. I think it is valuable to also quantify Hg mass, biomass times with Hg conc., not only with Hg concentrations, due to biomass also accumulate across the growing season.

*Reply:We agree with this suggestion. The main reason that we only valued the foliar Hg concentrations is to compare with the previous results of Hg concentrations in leaves (Laacouri et al., 2013; Moore et al., 1995; Obrist et al., 2021; Poissant et al., 2008; Wang et al., 2016; Zhang et al., 2009). We will consider valuing Hg mass in future research.*

Line 193. Due to different plants structure, perhaps Hg accumulated in foliage is not dominantly from atmospheric uptake as forests did, some foliage Hg may also from root uptake. At least some salt marsh studies have shown that Hg from roots can transport to marsh vegetation leaves due to higher Hg concentration contained in their roots (list as following). More studies are needed to demonstrate your conclusion here, such as belowground roots and rhizomes, and soils samples collect associate with Hg analysis.

such as studies from Cabrita et al., 2019 (Mercury mobility and effects in the salt-marsh plant Halimione portulacoides: Uptake, transport, and toxicity and tolerance mechanisms);

Weis and Weis, 2004 (Metal uptake, transport and release by wetland plants: implications for phytoremediation and restoration)).

*Reply:We agree with this suggestion. We have incorporated additional content in Page 3 (Line 56-65). The clarified text (Page 3, Line 56-65) reads as follows "Vegetation is generally considered a sink for atmospheric Hg, with the majority of Hg in vegetation leaves accumulated from the atmosphere (Jiskra et al., 2018; Obrist et al., 2017). Plant leaves accumulate Hg from the atmosphere mainly through stomatal uptake (Lindberg et al., 1992). Stamenkovic and Gustin (2009) suggested that the non-stomatal pathway of Hg deposition to the leaf cuticle and*

*subsequently retention and incorporation into leaf tissue also plays an important role in accumulating atmospheric Hg. Plant roots generally acted as a barrier of Hg transport from soils to shoots (Wang et al., 2015), and less than 10% of Hg in roots was transported to the aboveground portion of plants (Ericksen et al., 2003; Mao et al., 2013). Some studies also found that a great proportion of foliar Hg in Halophytes in salt marshes was translocated from the root (Canário et al., 2017; Cabrita et al., 2019; Weis and Weis 2004). The plausible reason is that plants in the hydroponic growth system have fewer apoplastic barriers (i.e. Casparian bands and suberin lamellae) in root architecture than plants grown in contaminated soils (Redjala et al., 2011)".*

*We also revised this sentence in the manuscript (Page 9, Line201-204) as follows "This result showed a clear pattern of continuous THg accumulation in foliage in boreal peatland plant species over time as has been shown for forests (Laacouri et al., 2013; Millhollen et al., 2006b; Rea et al., 2002), which can be attributed to foliar Hg accumulation from the air, given that plant roots act as a barrier of Hg transport from soils to shoots (Wang et al., 2015).".*

Line 211 Hg may also mobilize between plants, roots, rhizomes. Also other Hg sources perhaps, i.e. Hg conc in soils, uptake through roots, and then transport to the leaf may also contribute Hg conc increase.

**Reply:***We agree with this suggestion. The revised text (Page 10; Line 220-222) reads as follows "Although foliar Hg can transport to other plant organs, such as tree rings (Arnold et al., 2018; McLagan et al., 2022), and/or can be re-emitted into the atmosphere (Zheng et al., 2016; Yu et al., 2016; Yuan et al., 2019), the majority of foliar Hg by mass is generally incorporated into leaf tissue (Laacouri et al., 2013; Lodenius et al., 2003; Stamenkovic and Gustin, 2009). In addition, it is likely that less than 10% of Hg in roots was transported to the leaves (Ericksen et al., 2003; Mao et al., 2013).".*

---

## Author Response (AR2)

Dear Editor,

We are grateful for the opportunity to resubmit this manuscript for publication in Biogeosciences. Please find our response to all reviewers' and the editor's comments on the above manuscript. These comments were very helpful in clarifying some ideas and interpretations. In addition to those changes, we corrected a few typos and grammar which we identified during the review. We have addressed all of the comments below, to what we feel is a complete and thorough way. A detailed list of all relevant changes made in the manuscript is provided below.

Sincerely, Ting Sun and Brian A. Branfireun

**Additional changes to the manuscript**

These changes were not derived from the reviewers' comments, but from the authors' own assessment of the revised version after implementing the replies to those comments.

Line 185 Removed "the" in the sentence "Differences in foliage quality (%C, %N, and C:N) were analyzed using a multivariate ANOVA."

Line 221 Changed "decrease of" to "decrease in"

Line 275 Changed "decrease of" to "decrease in"

Line 314 Added the change percentage of foliar %C and %N after leaching of litterfall in Table 1.

Line 326 Changed the title "Quantity and characteristics of leachate dissolved organic matter" to "The roles of dissolved organic matter properties in Hg mobility during litter leaching phase".

**Author's response to the editor**

1. the response to reveiwer's concerns of " Line 121. For annual biomass, there is always variation between different years, depends on weather conditions, such wet or drought, warm or cool, etc." is not proper. please carefully read reveiwer's comment here and answer what he/she asked.

*Reply: We are grateful for the opportunity to clarify our response to the reviewers point. We agree that the annual vegetation biomass (and the relative distribution of species) is affected by weather conditions and will vary from year to year.*

*Interannual variability in precipitation and temperature will result in variability in plant biomass and distribution among plant species, however the dominant plant species distribution (as a fraction of total biomass) reported for one growing season by Palozzi and Lindo (2017) was sedges and sweet gale. The mean percent cover of few-seeded sedge, wire sedge, tussock sedge, and sweet gale from the sedge-dominated fen is $35.0 \pm 21.79\%$, $0.3 \pm 0.12\%$, $73.0 \pm 18.81\%$, $44.8 \pm 10.63\%$ (average $\pm$ standard error (SE)), respectively (Palozzi and Lindo 2017). We used this*

*information to justify the choice of species to consider for this study because of their high proportional contribution to overall biomass at this site.*

2. there is no data to quantify the atmospheric and root uptake.

*Reply: We did not investigate the uptake and translocation of Hg by these peatland plants in this study, because there is sufficient evidence in the literature to justify not doing so (i.e. leaf uptake is overwhelmingly important from a mass balance perspective) with rare exceptions (some evidence from halophytes; Canário et al., 2017; Cabrita et al., 2019; Weis and Weis 2004), the majority of of Hg in vegetation leaves accumulates from the atmosphere via surficial deposition and stomatal uptake of gaseous Hg (Jiskra et al., 2018; Obrist et al., 2017). Plant roots generally act as a barrier of Hg transport from soils to shoots (Wang et al., 2015). Although root uptake could be investigated under perhaps more controlled conditions and different objectives, we are confident that it would be a small proportion of Hg in leaves of these plants and would not have been a justifiable component of this study, which sought to evaluate the cumulative change in foliar Hg in these important wetland plant species. We have added a statement in the discussion (Lines204-205) to clarify that we don't know what proportion of Hg could be coming from other sources, and that this could be investigated further. The revised text (Lines204-205 ) reads as follows:"Further studies are needed to quantify the contribution of atmospheric and soil Hg to foliar Hg. "*

3. remove reference from conclusion section
*Reply: The reference has been removed from the conclusion section.*

4. for fig. 3, is the data enough to make a correlation analysis?
*Reply: Fifteen data points are enough to conduct a correlation analysis with the*

*acknowledgment that the relationship in this case is strongly leveraged by one species, which is very different from the other two. It assigns that Pearson correlation coefficient (r) is between − 1 and 1, where 0 is no correlation, 1 is total positive correlation, and − 1 is a total negative correlation. There is no strict requirement for the number of data, however as with any correlation analysis more data ensures a more robust analyses and confirmation that the data complies with the requirements of the test. We have added a statement in the discussion (Lines 284) to clarify that we are not drawing predictive conclusions from these regressions, but merely illustrating the coherence of the data across species. The revised text  (Line 284)  reads as follows:"More studies and data are needed to draw predictive conclusions."*

**Author's response to reviewer's comments (RC1)**

We are grateful for the reviewer's deeply informed and insightful comments. We have addressed the detailed comments that were made on the PDF copy of the earlier version of the manuscript. Please find our replies to your comments and suggestions below.

Line 23, you can change "." to ",".

**Reply:** *We agree with this suggestion. We have changed "." to "," in the manuscript (Line23).*

Line 34: where these Hg accumulated in plants come from?

**Reply:** *Please see our response to the editor's comment 2.*

Line 65 Here are more Hg accumulated studies in the peatland"

Grigal, D., Kolka, R., Fleck, J. & Nater, E. Mercury budget of an upland- peatland watershed. Biogeochemistry 50, 95–109 (2000).

Grigal, D. F. Mercury sequestration in forests and peatlands: A review. J. Environ. Qual. 32, 393–405 (2003)

Osterwalder, S. et al. Mercury evasion from a boreal peatland shortens the timeline for recovery from legacy pollution. Sci. Rep. 7, 16022 (2017).

Woerndle, G. E. et al. New insights on ecosystem mercury cycling revealed by stable isotopes of mercury in water flowing from a headwater peatland catchment. Environ. Sci. Technol. 52, 1854–1861 (2018).

*Reply: We have incorporated additional content in this section and cited more recent references. The revised text (Line 75-77) reads as follows:" Previous studies have found that the majority of Hg in plant leaves in wetlands was from the atmosphere (Brahmstedt et al., 2021; Enrico et al., 2016; Fay and Gustin 2007) and nonvascular plants (e.g., fungi, lichens, and mosses) had higher foliar Hg concentrations than vascular plants (Moore et al., 1995; Pech et al., 2022)."*

Line 118. Plot size sampling in 2018? Is that the same size with samples collected in 2019?

*Reply: Actually, the "plot" in 2018 should be "location". In 2018, five locations several hundred meters apart were selected in the sedge-dominated fen to serve as within-site replicates to account for potential local-scale variability. In 2019, seven 0.25 m$^2$ (0.5 m × 0.5 m) plots several hundred meters apart were selected at the end of August 2019 during senescence and before leaf off to estimate the annual biomass of senesced leaf. We have changed "plot" to "location" in the manuscript (Line 129).*

Line 121. For annual biomass, there is always variation between different years, depends on weather conditions, such wet or drought, warm or cool, etc.

*Reply: Please see our response to the editor's comment 1.*

Line 183. Meaning of F(1.73,24.26)and F(1.23,23.38)?

*Reply:Thank you for the reviewer's question. For the repeated-measures ANOVA, we get the results F(x,y) and p. F is the statistics of the repeated-measures ANOVA, p-value indicates that if the mean difference among all groups was statistically significant. x and y is calculated based on the degree of freedom between subjects and the degree of freedom within subjects. Based on Mauchly's Test of Sphericity, if the results did not follow Mauchly's Test of Sphericity, the results needed to be modified with epsilon (e). We double-checked our analysis throughout the manuscript and made the necessary corrections in the manuscript. The revised text (Line 192) reads as follows:" Foliar THg concentrations were related to time/leaf age ($F_{(3,36)}$ = 108.86, $p < 0.001$) and plant species ($F_{(2,12)}$ = 51.85, $p < 0.001$) (Fig. 1)."*

Line 189. I think it is valuable to also quantify Hg mass, biomass times with Hg conc., not only with Hg concentrations, due to biomass also accumulate across the growing season.

*Reply:We agree with this suggestion. The main reason that we only valued the foliar Hg concentrations is to compare with the previous results of Hg concentrations in leaves (Laacouri et al., 2013; Moore et al., 1995; Obrist et al., 2021; Poissant et al., 2008; Wang et al., 2016; Zhang et al., 2009). We will consider valuing Hg mass in future research.*

Line 193. Due to different plants structure, perhaps Hg accumulated in foliage is not dominantly from atmospheric uptake as forests did, some foliage Hg may also from root uptake. At least some salt marsh studies have shown that Hg from roots can transport to marsh vegetation leaves due to higher Hg concentration contained in their roots (list as following). More studies are needed to demonstrate your conclusion here, such as belowground roots and rhizomes, and soils samples collect associate with Hg analysis.

such as studies from Cabrita et al., 2019 (Mercury mobility and effects in the salt-marsh plant Halimione portulacoides: Uptake, transport, and toxicity and tolerance mechanisms);

Weis and Weis, 2004 (Metal uptake, transport and release by wetland plants: implications for phytoremediation and restoration)).

*Reply: Please see our response to the editor's comment 2. We have incorporated additional content in the manuscript (Line 56-65 and Line201-205 ) to clarify the sources of foliar Hg.*

Line 211 Hg may also mobilize between plants, roots, rhizomes. Also other Hg sources perhaps, i.e. Hg conc in soils, uptake through roots, and then transport to the leaf may also contribute Hg conc increase.

*Reply:We agree with this suggestion. Mercury can transport in plant organs (Arnold et al., 2018; McLagan et al., 2022) and can also be re-emitted to the air (Zheng et al., 2016; Yu et al., 2016; Yuan et al., 2019), however the majority of foliar Hg is generally incorporated into leaves (Ericksen et al., 2003; Mao et al., 2013). We have added this information to the manuscript. The revised text (Line 220-222) reads as follows:" Although foliar Hg can transport to other plant organs, such as tree rings (Arnold et al., 2018; McLagan et al., 2022), and/or can be re-emitted into the atmosphere (Zheng et al., 2016; Yu et al., 2016; Yuan et al., 2019), the majority of foliar Hg by mass is generally incorporated into leaf tissue (Laacouri et al., 2013; Lodenius et al., 2003; Stamenkovic and Gustin, 2009). In addition, it is likely that less than 10% of Hg in roots was transported to the leaves (Ericksen et al., 2003; Mao et al., 2013)."*

**Author's response to reviewer's comments (RC2)**

Sun et al. "Plant mercury accumulation and litter input to a Northern Sedge-dominated Peatland" investigated the foliar Hg concentration and flux via Sedge plant in peatland. In addition, they also carried out a leaching experiment to explore the Hg

behavior in leaching process driven by DOM. I think this MS is important research to understand the Hg biogeochemical cycle and well written. However, I think this discussion is not enough in this MS, especially in relationship between DOM and Hg. I hope more effective discussion should be added in the revised MS. Moreover, the figures should be revised fully.

*Reply: We are grateful for the reviewer's deeply informed and insightful comments. We have addressed the detailed comments that were made on the PDF copy of the earlier version of the manuscript. We also added more recent references. We think figures in black and white have complied with the journal's guidelines for figures. Please find our replies to your comments and suggestions below.*

*List of references that we added:*

[revised manuscript text omitted]

Introduction: It is too divergent. The author should focus on the Hg specific behavior in Hg cycles at peatland, highlighting the importance in global Hg cycles and differences from forest ecosystems. Add more recent references.

*Reply: We agree with this suggestion. We have modified the introduction and added more recent references based on the reviewer's suggestion. We added more information to clarify the sources of foliar Hg in the manuscript (Line 56-65), the role of boreal peatlands in global Hg cycling and the differences between forest and boreal peatlands in Hg cycling (Line 66-76). We clarified the importance of vascular plants in Hg cycling and why we choose boreal peatlands as the study site (Line 75-86). We added the role of dissolved organic matter in Hg mobility to clarify that The rapid and abundant leaching of DOM, especially those with higher aromaticity from litterfall may lead to large amounts of Hg leaching (Line 90-95). Please see the below details:*

*We added the information (Line 56-65) reads as follows: "Vegetation is generally considered a sink for atmospheric Hg, with the majority of Hg in vegetation leaves accumulated from the atmosphere (Jiskra et al., 2018; Obrist et al., 2017). Plant leaves accumulate Hg from the atmosphere mainly through stomatal uptake (Lindberg et al., 1992). Stamenkovic and Gustin (2009) suggested that the non-stomatal pathway of Hg deposition to the leaf cuticle and subsequently retention and incorporation into leaf tissue also plays an important role in accumulating atmospheric Hg. Plant roots are thought to generally act as a barrier of Hg transport from soils to shoots (Wang et al., 2015), and it has been shown that less than 10% of Hg in roots is transported to the aboveground portion of plants (Ericksen et al., 2003; Mao et al., 2013). Some studies have found that a great proportion of foliar Hg in halophytes in salt marshes was translocated from the root (Canário et al., 2017; Cabrita et al., 2019; Weis and Weis 2004). The plausible reason is that plants in the hydroponic growth system have fewer apoplastic barriers (i.e. Casparian bands and suberin lamellae) in root architecture than plants grown in contaminated soils (Redjala et al., 2011)."*

*We revised the text (Line 66-76) reads as follows:"Forest ecosystems are important sinks of atmospheric Hg and have received widespread attention from researchers (Risch et al., 2012; St. Louis et al., 2001; Wang et al., 2016; Zhang et al., 2009); however, studies about foliar Hg accumulation in other plant types in boreal peatlands ecosystems are few (see Moore et al., 1995) despite their critical role in the carbon (Gorham, 1991) and Hg cycles (Grigal, 2003). Boreal peatlands are a type of wetland that stores large amounts of Hg (Grigal 2003) and can be major MeHg sources to downstream ecosystems (Branfireun et al., 1996; Mitchell et al., 2008; St. Louis et al., 1994), given their anaerobic conditions, non-limiting amounts of inorganic Hg, and often available but limited amounts of sulfate (Blodau et al., 2007; Schmalenberger et al., 2007) and bioaccessible carbon facilitating net MeHg production (Mitchell et al., 2008). Elucidation of foliar Hg input from the dominant plant types to boreal peatlands is important to further estimate the supply of bioavailable Hg(II) for net MeHg production."*

*We added the information (Line 75-86) reads as follows: "Previous studies have found that the majority of Hg in plant leaves in wetlands was from the atmosphere (Brahmstedt et al., 2021; Enrico et al., 2016; Fay and Gustin 2007) and nonvascular plants (e.g., fungi, lichens, and mosses) had higher foliar Hg concentrations than vascular plants (Moore et al., 1995; Pech et al., 2022). Although foliar Hg concentration is lower in vascular plants than in nonvascular plants, Hg mass input to peatlands may be substantial, given the greater litter input from vascular plants than from nonvascular plants (Frolking et al., 2001). With more bioaccessible litter and leachate than bryophytes (Hobbie, 1996; Lyons and Lindo, 2019), vascular plant inputs also have a faster initial decomposition rate (0.2 y-1) than bryophytes (0.05-0.08 y-1) (Frolking et al., 2001), leading to a rapid Hg release to the soil Boreal peatlands are experiencing rising temperatures due to climate change (IPCC, 2018) that is likely to both increase aboveground biomass in vascular plant-dominated peatlands (Tian et al., 2020) and promote a shift from moss-dominated to more vascular plant-dominated plant communities (Buttler et al., 2015; Dieleman et al.,*

*2015; Weltzin et al., 2000) further affecting Hg deposition (Zhang et al., 2016). To date, the amount of atmospheric Hg accumulated in dominant plants in the vascular plant-dominated (i.e., graminoid plants and shrubs) peatlands, an important type of boreal wetlands (Rydin and Jeglum, 2013), is unknown.”*

*We added the information (Line 90-95) reads as follows:“It has been established that dissolved organic matter (DOM) is closely related to Hg mobility in terrestrial and aquatic ecosystems (Haitzer et al., 2002; Ravichandran, 2004; Kneer et al., 2020), given the strong affinity between Hg and reduced sulfur groups (i.e., thiols) in DOM (Xia et al., 1999). DOM with higher aromaticity has more thiols ligands and has a stronger correlation with Hg (Dittman et al., 2009). The rapid and abundant leaching of DOM, especially those with higher aromaticity from litterfall may lead to large amounts of Hg leaching.*

*We added the information (Line 97-99) reads as follows:“Despite previous studies showing that Hg mass in live leaf leachate is insignificant compared to that on leaf surfaces and in SOM (Rea et al., 2001; Rea et al., 2000), litterfall generally lacks structural integrity and likely leaches more Hg compared to live leaves.”*

Line 23: While should be revised.

**Reply:** *We have revised this in the manuscript (Line23).*

Line 27: It is inconsistent.

**Reply:** *We agree with this comment. The clarified text (Line 27) reads as follows “Foliar Hg concentrations decreased early in the growing season due to growth dilution, and after that were subsequently positively correlated with leaf age (time)”.*

Line 70: It is confused. Further explain it.

*Reply: We agree with this suggestion. The clarified text (Line 77-79) reads as follows "Although foliar Hg concentration is lower in vascular plants than in nonvascular plants, Hg mass input to peatland might be important, given the more amount of litter input from vascular plants than from nonvascular plants (Frolking et al., 2001)."*

Line 71-72: Plant decomposition turnover time should be mentioned as you talked about the Hg biogeochemical cycles.

*Reply: We agree with this suggestion. We have incorporated additional content in this section. The revised text (Line 79-81) reads as follows "With more bioaccessible litter and leachate than bryophytes (Hobbie, 1996; Lyons and Lindo, 2019), vascular plant inputs also have a faster initial decompose rate ($0.2\ y^{-1}$) than bryophytes ($0.05\text{-}0.08\ y^{-1}$) (Frolking et al., 2001), leading to a rapid Hg release to the soil".*

Line 73-83: I cannot get the significance of this section. I agreed with those points, but global warming has nothing to do with this research. Only one sentence is enough to highlight the importance of peatlands.

*Reply:We agree that one sentence is enough for this section. The clarified text (Page 4, Line 81-87) reads as follows "Boreal peatlands are experiencing rising temperatures due to climate change (IPCC, 2018) which is likely to both increase aboveground biomass in vascular plant-dominated peatlands (Tian et al., 2020) and promote a shift from moss-dominated to more vascular plant-dominated plant communities (Buttler et al., 2015; Dieleman et al., 2015; Weltzin et al., 2000) further affecting Hg deposition (Zhang et al., 2016). To date, the amount of atmospheric Hg accumulated in dominant plants in the vascular plant-dominated (i.e., graminoid plants and shrubs) peatlands, an important type of boreal wetlands (Rydin and Jeglum, 2013), is unknown."*

Line 105: Offer the percentage of each dominate species.

*Reply: We have incorporated the species percent cover of each dominant species in*

*this section. The revised text (Line 115-117)reads as follows "The mean species percent cover of few-seeded sedge, wire sedge, tussock sedge, and sweet gale from the sedge-dominated fen is 35.0 ± 21.79%, 0.3 ± 0.12%, 73.0 ± 18.81%, 44.8 ± 10.63% (average ± standard error (SE)), respectively (Palozzi and Lindo 2017)."*

Line 130 and SI 32-37: Why did you choose the DORM-4 (Fish protein certified reference material) as the standard sample, not the plant standard samples?

*Reply:Thank you for the reviewer's comment. In this study, DORM-4 was used to validate instrument (Milestone™ DMA-80) recovery and stability not for the method validation step (accuracy, precision, recovery of samples). Milestone™ DMA-80 is very stable and reliable. A single calibration suits all wide variety of samples with concentrations ranging from ppm to ppt. The DMA-80 calibration provides long-term reliability due to the stability of the system and the long lifetime of the catalyst tube and gold amalgamator. These features allow us to eliminate the daily calibrations often required by conventional instrumentation. In this study, the concentration range of the regular calibration was from 0 to 1 mg/kg. Total Hg concentration in DORM-4 is 0.410 ± 0.055 mg/kg, which is in the middle of the calibration concentration range and thus can validate instrument recovery and stability very well. There is one CRM "IAEA-140/TM Trace elements and methylmercury in seaweed"for total mercury concentration, but the Hg concentration of CRM is 0.038 (0.032-0.044 mg/kg), which is not suitable to validate instrument recovery and stability.*

Line 183: Explain the subscript in F(1.73,24.26).

*Reply: Please see our response to the reviewer's comments "Line 183. Meaning of F(1.73,24.26)and F(1.23,23.38)?"*

Line 208-209: In references, the Hg concentration in foliage showed the linear increasing, inconsistent with the decreased uptake rate as description of the sentence.

*Reply:Thank you for this comment. The rate of leaf Hg uptake means the flux rate of*

*gaseous Hg to plant, which is different from foliar Hg concentration. Despite the decrease in foliar Hg uptake rate, the leaf still continuously uptake Hg from the atmosphere and the majority of Hg was incorporated into leaves, thus, the Hg concentration in foliage still showed a linear increase.*

Line 218: The larger leaf also caused the bigger biomass, offsets the stomates effect. How to explain it?

*Reply: We agree with the reviewer's point that "bigger leaf not only has more stomatal openings and more surface but also has more biomass.". The more biomass may offset effects of stomates on atmospheric Hg accumulation by leaves to a certain degree. Leaf biomass may do not proportionally increase with leaf area and stomates, leading to a higher absolute Hg concentration in tussock sedge leaves than in few-seeded sedge/wire sedge leaves.The clarified text (Line229-234) reads as follows "A larger leaf has a higher density of stomate and thus more leaf accumulation of atmospheric Hg (Laacouri et al., 2013; Millhollen et al., 2006; Stamenkovic and Gustin, 2009). A larger leaf area may also provide more adsorption sites for non-stomatal Hg uptake. Increased biomass corresponding with a bigger leaf area can offset the effects of stomate number on atmospheric Hg accumulation by leaves to a certain degree. A plausible explanation is that leaf biomass does not proportionally increase with leaf area and stomata, leading to a higher absolute Hg concentration in tussock sedge leaves than in few-seeded segde/wire sedge leaves.".*

Line 229-231: It is insipid. Further explain is needed to clarify the reason why the Hg concentration in peatland vegetation is lower than that in tree litter.

*Reply:We agree with this suggestion. We have incorporated additional content in this section. The revised text (Page 10-11, Line 239-255) reads as follows "Concentrations of Hg in senesced leaves of few-seeded sedge/wire sedge, tussock sedge, and sweet gale (6.58 ng g-1 to 12.77 ng g-1) were lower than that reported in tree litter (21 ng g-1 – 78 ng g-1) in North-America and Europe (Laacouri et al., 2013; Obrist et al., 2021;*

*Poissant et al., 2008; Rea et al., 2002; Wang et al., 2016) but similar to that previously reported for grasses and herbaceous plants (~10 ng g-1) (Moore et al., 1995; Olson et al., 2019). The foliar Hg concentrations for plant species in this study increased 1.3-2.0 times over the growing season, which was smaller than that (3-11 fold) reported for trees (Laacouri et al., 2013; Poissant et al., 2008; Rea et al., 2002). The above results further confirm that foliar Hg concentrations differ among vegetation types (Demers et al., 2007; Moore et al., 1995; Obrist et al., 2012; Richardson and Friedland, 2015). It has been suggested that Hg previously retained in leaves can be photo reduced to Hg0 that is re-emitted to the atmosphere, and consistent Hg0 re-emission from the foliage is positively related to photosynthetically active radiation (PAR) (Yuan et al., 2019). The plants in open boreal peatlands lacking a tree overstorey like that in this study would receive very high exposure to ultraviolet (UV), which may result in a greater photoreduction of Hg previously retained in leaves and then Hg loss than tree leaves that are more often shaded. Moreover, despite angiosperms having higher stomatal conductance due to fewer stomata but more numbers (de Boer et al., 2016; Jordan et al., 2015), stomatal opening in dark-adapted leaves after light exposure was generally faster in gymnosperms than in angiosperms but stomatal closing upon the darkness of light-adapted leaves was faster in angiosperms than in gymnosperms (Xiong et al., 2018). This phenomenon may lead to a higher Hg concentration in trees (a type of gymnosperms) than in sedges and sweet gales (two types of angiosperms). More studies are needed to elucidate this mechanism of foliar Hg accumulation by different plant types.".*

Line 334-342: This discussion is not enough. What does the SUA represent? I hope not only the amount of aromaticity. You should explain more about each factor. For example, "indicating that leached DOM from tussock sedge and few-seeded sedge/wire sedge leaves had higher aromaticity and less bioaccessible than that from the sweet gale leaves", the higher aromaticity and less bioaccessible, so what? This discussion is meaningless.

*Reply: DOM characteristics (i.e. bioaccessibility) are presented by $SUVA_{254}$ and fluorescence indices in this study. $SUVA_{254}$ values indicate the amount of aromaticity of DOM, where lower $SUVA_{254}$ indicates lower DOM aromaticity and relatively greater bioaccessibility. Higher FI and BIX and lower $HIX_{EM}$ values indicate DOM is relatively more bioaccessible. The explanation of $SUVA_{254}$, FI and BIX was already given in the original manuscript (Line 167-178). We have incorporated additional content in the discussion.*

*The revised text (Line 344-358) reads as follows "Characteristics of DOM also varied among plant species ($SUVA_{254}$: $F_{(2,42)} = 24.02$, $p < 0.001$; FI: $F_{(2,42)} = 11.24$, $p < 0.001$; BIX: $F_{(2,42)} = 125.48$, $p < 0.001$) (Fig. 6 and Table 2). Based on post hoc tests, there were significant differences in SUVA254 and FI between sweet gale and sedges (few-seeded sedge/wire sedge) only and BIX among all plant species. FI (1.2-1.8) and BIX (<1.0) reflected that DOM in leachate was generally of plant origin, suggesting that the microbially-derived OM was a smaller component. The mean value of $SUVA_{254}$ in leachate followed the sequence: tussock sedge ≈ few-seeded sedge/wire sedge > sweet gale leaves, respectively, indicating that leached DOM from tussock sedge and few-seeded sedge/wire sedge leaves had higher aromaticity and higher molecular weights than that from the sweet gale leaves. $SUVA_{254}$ was negatively related to DOM concentrations ($F_{(1,43)} = 48.37$, $p < 0.001$, $y = -0.69x + 3.93$, $R^2 = 0.53$) when all plant species were considered, suggesting that sweet gale prefers to release more amount of lower aromatic DOM.*

*Previous studies have found that characteristics of DOM controlled Hg mobility and methylation (Cui et al., 2022; Jiang et al., 2018; Ravichandran 2004; Xin et al., 2022; Wang et al., 2022). Hg is tightly and readily bound to reduced sulfur groups (i.e., thiols) in DOM (Ravichandran, 2004; Xia et al., 1999). Mercury weakly binds to carboxyl and phenol functional groups in DOM after all thiol groups are occupied at relatively high Hg concentrations (Drexel et al., 2002; Graham et al., 2012), which is atypical in most natural environments in which Hg concentrations are relatively low. Higher terrestrial (plant-derived) DOM had a greater DOM-Hg affinity (Wang et al.,*

*2022). Additionally, DOM with higher aromaticity and molecular weight strongly bonded with Hg(II), potentially because these DOM provide more sulfidic groups such as thiols (Dittman et al., 2009; Wang et al., 2022). Therefore, terrestrial DOM and/or DOM with higher aromaticity and molecular weight may transport more Hg into peat soils during the litter leaching phase." and "The concentrations of soluble THgaq were significantly related to SUVA$_{254}$ values ($F_{(1,41)}$ = 52.06, p < 0.001, y = 0.09x – 0.09, $R^2$ = 0.55; Fig. 7). This result suggested that DOM with higher aromaticity plays an important role in controlling Hg mobility (Ravichandran, 2004). The value of $R^2$ was only 0.55, which can be attributed that the number of reduced sulfur groups in DOM far exceeds the amount of Hg in natural environments and other factors, such as pH and sulfide may affect the binding between DOM and Hg (Ravichandran, 2004). In this study, DOM with higher aromaticity may transport more Hg from litter to soils, and senesced leaves of sedges had a higher potential in leaching Hg into peatland soils than the senesced leaves of sweet gales in this study.".*

Line 342-343: I can find the evidence to support that stimulate biological degradation and Hg methylation in this study.

***Reply:****We agree with this suggestion. There are many papers about the bioaccessible DOM (e.g., organic acids, sugars, amino acids) can stimulate biological degradation (Ganjegunte et al., 2006; Zhang et al., 2019) and Hg methylation (Mitchell et al., 2008; Schaefer et al., 2011; Leclerc et al., 2015). (See the below references). However, we removed this information from the manuscript to make the section focusing on Hg cycling in boreal peatlands.*

*Ganjegunte, G. K., Condron, L. M., Clinton, P. W., Davis, M. R., Mahieu, N.: Effects of the addition of forest floor extracts on soil carbon dioxide efflux.Biology and Fertility of Soils, 43, 199-207, DOI10.1007/s00374-006-0093-6, 2006.*

*Zhang, Z. Y., Wang, W. F., Qi, J. X., Zhang, H. Y., Tao, F., Zhang, R. D.: Priming effect of soil organic matter decomposition with addition of different carbon substrates. Journal of Soils and Sediments, 19, 1171-1178. DOI10.1007/s11368-018-2103-3. 2019*

Line 381-382: In this MS, I cannot find any MeHg data. But the authors always highlighted the MeHg production. This is confused.

*Reply: We agree with this suggestion. The clarified text (Page 21, Line 406 ) reads as follows "The THg concentrations in senesced leaves in this study are relatively lower than that in the forest litterfall".*

Line 383: This conclusion is not available in this MS.

*Reply: We agree with this suggestion. We have removed this sentence from the manuscript.*